# Reproducibility Report: Rethinking Softmax Cross-Entropy Loss for Adversarial Robustness

## Reproducibility Summary

.

### Scope of Reproducibility

Pang et.al. [1] presented Max-Mahalanobis center (MMC) loss and argued that MMC loss is adversarial more robust than SCE. The author's SCE loss conveys inappropriate supervisory signals to the model, leading to sparse sample density in the feature space. In this reproducibility challenge we verify the claims that training with MMC loss produces adversarially robust models while also enabling accuracy comparably with models trained with SCE loss.

### Methodology

We used the code as present in the repository provided by Pang et.al. [1]. We used their files to implement our experiments and test their hypothesis. We used Nvidia GeForce RTX 2080 Ti to perform all our experiments. It took a total of around 500 GPU hours.

We used adaptive attacks to test out the main claims of the paper. Our main goal was to prove the various hypothesis stated by the authors.

We also reproduce the MMC loss and optimal center generation algorithm in the PyTorch framework, which can help the PyTorch practitioner facilitate further research

### Results

We reproduced all the experiments as done by Pang et.al. [1] and could not see significant difference between the our results. All the results were within 2% of the values presented in the paper. We could also validate the hypothesis as stated by the authors of the paper. We believe that the paper gives a very good idea of what other objectives other than SCE loss could look like.

### What was easy

It is easy to replicate the originals results because the code was publicly available. Also implementing MMC loss was also pretty straight forward.

### What was difficult

The paper is very theoretical and it was difficult to understand some parts of it. Additionally, running adaptive attacks was tough because you had to go and change the loss function in cleverhans library for every experiment that had to be run. This was a tedious task. There were some places where we had to look at the proper documentation of a library to understand what was actually happening in the code.

### Communication with original authors

Some of our doubt regarding the theory and implementation details were clarified by the original authors via email and in the issues of their Github repository.

Submitted to ML Reproducibility Challenge 2020. Do not distribute.

## 1  Introduction

Deep Neural Networks have shown great success in many vision and language-related tasks. However, Deep neural networks (DNN's) are vulnerable to small input perturbations that are indistinguishable to the human eye but can easily fool the neural network, as demonstrated in [2, 3]. These perturbed inputs are known as adversarial attacks, and these attacks drive the trained models to classify objects which were previously classified with high accuracy wrongly. This unexpected behavior of DNNs raises some of the security concerns in DNN based systems [4, 5] thus limits the usage of DNN's in self-driving cars, robotics, and other related fields. The existence of adversarial nature in deep neural networks is still an open problem, and this has led to a plethora of publications on adversarial attacks and robustness.

Several methods exist for achieving adversarial robustness, such as [6, 7] proposes verification-based methods and training provable robust network. The only problem with these verification based methods is that they are slow and hard to scale. Other adversarial defense method includes adversarial training (AT) [8] of networks. These methods have shown state-of-the-art performance; however, AT is usually accompanied by a drop in accuracy on clean inputs, and AT is computationally expensive, as demonstrated in our experiments.

The original paper [1] introduces a novel MMC loss objective that significantly increases the robustness against strong adversarial attacks with little additional computation as compared to SCE loss. The paper presents the theoretical shortcoming of SCE loss function in inducing high sample density regions in the feature space. The MMC loss function introduces untrainable class centers around which the sample gathers in the feature space by minimizing the squared norm between the data points and the corresponding class center. These untrainable class centers are at an optimal distance from each other. The authors also investigated the theoretical foundations of MMC loss and demonstrated how higher density regions are induced around the class centers using the MMC loss compared to SCE loss. Our main contribution is listed below -

- We reproduce the results mentioned in the original paper and validate the results presented in the original paper.
- We further perform experiments to validate claims and assumptions made in the original paper. These experiments conclude that MMC loss can be used as a reliable metric of uncertainty on predictions and demonstrates substantial robustness to strong adaptive attacks. Additional experiments includes training time comparsion and effect of optimizers on MMC loss.
- Finally, we re-implement the optimum center generation algorithm (initially present in MATLAB ".mat" file) and MMC loss in Pytorch to facilitate further research in this area. We then present demerits of MMC loss over SCE. We also implement Hierarchical Max-Mahalanobis(HMMC) a variant of original MMC loss.

**Outline of the paper:** The immediate next section 2 presents a detailed discussion on theory related to sample density induced by SCE loss and MMC loss. Section 3 presents our experimental setup in reproducing our results. 4 contains the results of our experiments and a discussion on the results.

## 2  Theory

In this section, we will present the theoretical foundation related to MMC loss and SCE loss. Firstly, we mathematically define the induced sample density then we compare the relative induced sample density in the feature space for SCE loss and MMC loss.

### 2.1  Sample Density

The sample density [9] $SD(z)$, is defined as -

$$SD(z) = \frac{\Delta N}{\text{VOL}(\Delta B)}$$

where z is defined as a point in the feature space $z = Z(x)$, corresponding to an input $x$ in the dataset $D$ having $N$ number of training samples. Vol() denotes the volume of a set, $\Delta B$ represents the small neighboring region near the point $z$, and $\Delta N$ denotes the number of training points in $\Delta B$. Note that the mapped feature $z$ still corresponds to a particular label y.

**Remark 1:** The distribution of the samples in the feature space is directly influenced by loss $L(z, y)$ used during training. Since the supervisory signal is loss minimization, the sample density mainly varies along the orthogonal to loss contours.

As a consequence of *Remark 1* we can define $\Delta B$ in feature space as $\Delta B = \{z \in R^d | L(z, y) \in [C, C + \Delta(C)]\}$, where $C = L(z, y)$, and $\Delta C > 0$ is a small value. Now we can define $Vol(\Delta B)$ to be equal to the volume between the loss contours $C$ and $C + \Delta C$ for a label $y$ in the feature space.

## 2.2 Generalized SCE Loss

The family of SCE loss and its variants can be defined as -

$$L_{g-SCE}(Z(x), y) = -1_y^T log[softmax(h)]$$

where the logit $h = H(z) \in R^L$ is a general transformation of the feature $z$. The linear transformation $h = Wz + b$ is usually used in conjunction with SCE loss. A couple of other transformation has also been proposed, [10] proposed use of large-margin Gaussian Mixture loss, where $h = -(z - \mu_i)^T \Sigma(z - \mu_i) - m\delta_{i,y}$. [11] proposed the Max-Mahalanobis linear discriminant analysis, where $h_i = -||z - \mu_i^*||_2^2$. The authors argue that all the logits fall under the family of g-SCE loss with quadratic logits:

$$h_i = -(z - \mu_i)^T \Sigma_i(z - \mu_i) + B_i$$

where $B_i$ are the bias variables. Also, linear transformation is a special case of the quadratic logits.

The authors proved that, the sample density induced by g-SCE loss is proportional to $N_{k,\tilde{k}}$. $D_{k,\tilde{k}}$ refers to the total set of data points in the dataset whose true class is $k$ while $\tilde{k}$ is the class with the highest prediction amongst other classes. $N_{k,\tilde{k}}$ is just the total number of points in $D_{k,\tilde{k}}$. Formally given $(x, y) \in D_{k,\tilde{k}}$, $z = Z(x)$ and $L_{g-SCE}(z, y) = C$, the sample density near the feature point $z$ is

$$SD(z) \propto \frac{N_{k,\tilde{k}} \cdot p_{k,\tilde{k}(C)}}{[B_{k,\tilde{k}} + \frac{log(C_e - 1)}{\sigma_k - \sigma_{\tilde{k}}}]^{\frac{d-1}{2}}}$$

where for the input-label pair in $D_{k,\tilde{k}}$, $L_{g-SCE} \sim p_{k,\tilde{k}}(C)$.

**Remark 2:** The author argued that the problem in SCE loss mainly roots from applying the softmax function in during the training procedure. Softmax function causes the loss value to depend only on the relative relation among logits. This dependency leads to indirect and unexpected supervisory signals on the learned features representation, such that the points with low loss values tend to spread over the space in an sparsely. The authors also argued that in practice, the feature points do not move to infinity due of the existence of batch normalization layers in DNNs.

**Remark 3:** While deriving the loss contours induced by g-SCE loss an assumption is taken that $log[\Sigma_{l \neq y} exp(h_l)]$ can be approximated by $h_{\tilde{y}}$ where $h_{\tilde{y}} = argmax_{l \neq y} h_l$. However we found that it is not always true as demonstrated in our experiments ref section 4.

**Remark 4:** Through simple derivation, the authors proved that the loss contour induced by g-SCE loss is $(d - 1)$ dimensional hypersphere.

## 2.3 MMC Loss

The **Max-Mahalanobis Center (MMC)** loss is defined as -

$$L_{MMC}(Z(x), y) = \frac{1}{2}||z - \mu_y^*||_2^2$$

where $\mu^* = \{\mu_l^*\}_{l \in [L]}$ are the centers of the Max-Mahalanobis distribution (MMD) [12]. MMD is a special case of gaussian mixture distribution with an identity covariance matrix. The MMD centers $\mu^*$ are a set of untrainable centers calculated before the training procedure. These centers act as a converging point of all the training samples belonging to a specific class $y$. Another interesting point to note about MMC loss is that it is defined in regression format without softmax activation.

**Remark 5:** The MMC loss centers are untrainable and fixed at the starting of the training process. These centers are located such that the minimum angle between any two centers is maximized. For example, in the case of 2 centers, they will be on a line; in the case of 3 centers, they will be present at the vertices of an equilateral triangle, and for four centers, they will be positioned on the vertices of a regular tetrahedron as shown in Figure 1.

In the original paper [1] the author proved that, the sample density induced by the MMC loss is proportional to $N_k$ rather than $N_{k,\tilde{k}}$ as in the case of SCE loss and its variants. Formally given $(x, y) \in D_k$, $z = Z(x)$ and $L_{MMC}(z, y) = C$, the sample density near the feature point $z$ is

$$SD(z) \propto \frac{N_k \cdot p_k(C)}{C^{\frac{d-1}{2}}}$$

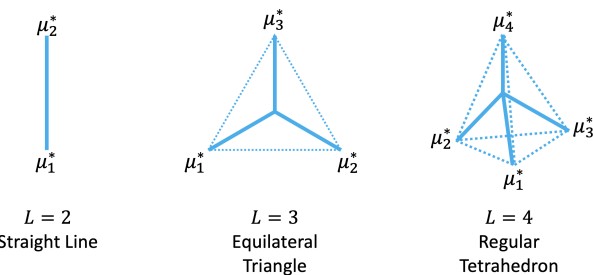

Figure 1: MMC centers

where for the input-label pair in $D_k$ there is $L_{MMC} \sim p_k(C)$.

Now we list down some of merits of using MMC loss over SCE loss -

- **Higher sample density:** As mentioned above, the sample density induced by MMC loss is proportional to $N_k$, where $N_k$ on an average is $N/num\_class$. While on the other hand, the g-SCE loss is proportional to $N_{k,\tilde{k}}$ and on average, it is equal to $N/(num\_class)^2$.

- **Faster Convergence:** The MMC loss explicitly focuses on minimizing the intra-class distance between the training samples and fixed class centers. There is no internal trade-off between intraclass dispersion and inter-class dispersion, which leads to faster convergence.

- **Uncertainty Estimation:** The MMC loss induces high-density regions in the feature space. Thus any sample having a feature representation this is not close to the any of the class center makes it very likely that it does not belong to any of the respective classes and hence MMC loss can also be used as a reliable metric to measure the uncertainty on the predictions.

## 2.4 Scope of reproducibility

The paper talks about better utilising the dataset to train adversarially robust models while also not losing out the high accuracies. They aim to do this with a new loss function Max Mahalanobis Centre Loss. They first show that the Softmax Cross-Entropy loss(SCE) conveys supervisor inappropriate supervisory signals to the model, leading to sparse sample density in the feature space, and demonstrates how MMC loss is not afflicted with the same problem.

This section roughly tells a reader what to expect in the rest of the report. Clearly itemize the claims you are testing:

- Reliable robustness even under strong adaptive attacks.
- MMC loss value also becomes a reliable metric of the uncertainty on returned predictions.
- MMC Loss is not much computationally expensive than SCE loss.
- Higher sample density is induced by MMC loss as compared to SCE loss.
- Global and Feature feature representation comparison between SCE and MMC loss.

Each experiment in Section 4 will support (at least) one of these claims, so a reader of your report should be able to separately understand the *claims* and the *evidence* that supports them.

## 3 Methodology

In this section, we will describe all the experimental settings involved during the training and inference phase. We will also report the adversarial attacks used to evaluate the performance.

For our experiments, we have used MNIST [13], CIFAR-10 and CIFAR-100 [14] and datasets to evaluate the performance of the proposed losses. The paper uses momentum **SGD(momentum value - 0.9)**with every model they have trained. All MNIST based-models have been trained for 50 epochs, and all CIFAR-10 and CIFAR-100 based-models have been trained for 200 epochs. The learning rate is initially 0.01 but decays by a factor of 0.1 at both 100 and 150 epochs. We used Nvidia GeForce RTX 2080 Ti to perform all our experiments. The network architecture used as a backbone is ResNet-32 with five core layer blocks[15]. This architecture has been kept constant in all of our experiments.

The original code is present at  and we release our code at  .We use the same set of hyperparameters that have been used by the authors to maintain consistency with the paper that we are reproducing. However, we found several minor inconsistencies while replicating the results using the original code-base. These are as follows -

- Use of Data-Augmentation Methods: The authors have used a few data augmentation methods like flipping (horizontal flip), shifting (both vertical=0.1 and horizontal=0.1), and ZCA whitening ($\epsilon=10^{-6}$), which has not beeen mentioned in the original paper.
- MMC-10(rand) has not been implemented: The author has given flags for training using MMC loss with random centers, but it has not been implemented in the repository.

## 3.1 Adversarial attacks

We are using the same adversarial attack models as used in the Pang [1]. The attacks they have used are comprehensive and cover many threat models, giving a better evaluation of the proposed loss.

- **White-box $l_\infty$ distortion attack:** The PGD method proposed by Madry [16] has been widely studied and is said to be a universal first order adversary.
- **White-box $l_2$ distortion attack:** Carlini and Wagner (C&W) [17] attacks are used in the paper.
- **Black-box transfer-based attack:** Momentum Iterative Method (MIM) [18] is used.
- **Black-box gradient-free attack:** SPSA attacks are used.
- **General-purpose attack:** The paper also evaluates the robustness of the model to addition of random noises [19] and random rotation [20].

All the above attacks are modified to be adaptive attacks [21, 17] to remove the effect of gradient masking while evaluating the robustness of the proposed loss.

# 4 Results and Discussion

In this section, we present the results of our experiment and discussion based on our observations.

## 4.1 Evaluation on Adversarial Attacks

We first evaluate the models trained on MMC loss and SCE loss with different adversarial attacks as proposed in the original paper [1]. We were able to reproduce all the adversarial attacks mentioned in the paper, and no inconsistencies related to accuracy were found. So, we think that putting the two tables here would provide little value to the reader. The models have been evaluated both on adaptive attacks and non-adaptive attacks. We observed that across all training methods involving MMC loss, testing accuracy under non-adaptive untargeted attacks is always significantly greater than adaptive untargeted attacks. We observe that methods trained with either untargeted or targeted attacks show greater accuracy under adaptive targeted PGD attacks than under non-adaptive targeted PGD attacks for perturbations of ($\epsilon = \frac{8}{256}$) while the reverse is true for PGD attacks with perturbation of ($\epsilon = \frac{16}{256}$).

## 4.2 Training Time Comparison

We also evaluate our model based on the time taken for the training procedure keeping the epochs and batch size fixed across the dataset. We train our model for 50 epochs with a batch size of 50.

| Method | Dataset | Number of classes | Timing (in min) |
|---|---|---|---|
| Standard SCE | MNIST | 10 | 39.8 |
| Standard MMC | MNIST | 10 | 48.075 |
| Adversarial SCE | MNIST | 10 | 271.14 |
| Adversarial MMC | MNIST | 10 | 280 |
| Standard SCE | CIFAR10 | 10 | 54.04 |
| Standard MMC | CIFAR10 | 10 | 55.23 |
| Standard SCE | CIFAR100 | 100 | 42.8 |
| Standard MMC | CIFAR100 | 100 | 45.55 |

Table 1: Training time for different methods

The difference between timings of SCE and MMC loss is shown in Table 1. Little difference is observed between training times for MMC loss and SCE loss. This observation validates the claim of the paper that MMC is not much computation expensive than SCE loss. We also observe that (AT) hugely increases the training time, nearly $\approx 7\times$ the training time under standard training procedure for both SCE and MMC loss.

## 4.3 Effects of Optimizer

We try different optimizers to check which optimizer is suitable with MMC loss. As you can see in Table 2, Momentum SGD gives the best accuracy with the least time .

| Optimizer | Resulting accuracy | Training Time(min) |
|---|---|---|
| Momentum SGD | 99.69% | 47.7 |
| Adam | 99.53% | 62.7 |
| RMSProp | 99.40% | 55.9 |

Table 2: Experiment to determine optimal $C_{MM}$ value for use with MMC loss. Models trained on MNIST dataset for 50 epochs using MMC loss.

## 4.4 Uncertainty Prediction

We validate one of the merits of MMC, that is, MMC can reliably estimate the uncertainty in the prediction. To validate this hypothesis, we tested the performance of the models with a random image. This hypothesis will test the confidence aware learning capability of MMC loss over SCE loss. E.g., we are interested in knowing the output score of SCE and MMC loss when we feed a car image as a test image into a model trained on a cat and dog dataset. If the model gives high probability scores to a particular class, it implies that the model is overly confident about the predictions. Ideally, it should be around 0.5 for dog class and 0.5 for cat class for SCE loss.

For this experiment, an image of a lion resized to $(32 \times 32)$ was taken for the models trained on MNIST, and for models trained on CIFAR-10 and CIFAR-100, a random image from the MNIST dataset was used. The scores of the final layers are given in Table 3. The results demonstrate the uncertainty of prediction when MMC loss is used. Models with SCE loss give high probabilities for even irrelevant classes, which is undesirable. MMC loss gives a high norm value for all classes, which implies that the lion's feature representation is very different from the class center representation and is nearly equidistant from all the class centers. This experiment demonstrates another hypothesis by the authors: MMC loss value also becomes a reliable metric of the uncertainty on returned predictions. In Table 3 the values of the top three classes have been reported.

| Method | Dataset | Class 1 | Class 2 | Class 3 |
|---|---|---|---|---|
| Standard SCE | MNIST | 0.997 | 0.00292 | 0.00004 |
| Standard MMC | MNIST | -1618.7 | -1670 | -1680 |
| Standard SCE | CIFAR-10 | 1 | 0 | 0 |
| Standard MMC | CIFAR-10 | -10143 | -10152 | -10564 |
| Standard SCE | CIFAR-100 | 1 | 0 | 0 |
| Standard MMC | CIFAR-100 | -138439 | -139005 | -141384 |

Table 3: Top 3 Scores on different models. Models trained on MMC loss give the distance from each class center as scores while models trained on SCE loss give probabilities for each class as scores.

## 4.5 Relative Dependency in g-SCE Loss

We also verify the assumption that $log[\Sigma_{l\neq y}exp(h_l)]$ cannot be smoothly approximated by $h_{\tilde{y}}$ where $h_{\tilde{y}} = argmax_{l\neq y}h_l$ on each datapoint as mentioned in *Remark 3*. For this experiment we use ResNet-32 model trained on CIFAR10 dataset using SCE loss. We then compare $(h_{\tilde{y}})$ where $h_{\tilde{y}} = argmax_{l\neq y}h_l$ with remaining eight class score values. Our observation is that for CIFAR10 dataset around 9.16% of the samples have comparable ($< 5\times$) second and third class scores. Thus $log[\Sigma_{l\neq y}exp(h_l)]$ cannot be approximated by just $h_{\tilde{y}}$. For more accurate results it is advisable to use top five values of the class scores ($h_l$ where l is class centers having top 5 scores such that $l \neq y$) for a clean approximation.

### 4.6 Effect of MMC loss Constant ($C_{MM}$)

We also investigate a model's performance trained with MMC for different scaling constant ($C_{MM}$). For our experimentation, we used the value of $C_{MM} = \{1, 5, 10, 100\}$ and measured the effect of ($C_{MM}$) on the accuracy of the model. As clear from Table ($C_{MM} = 10$) yields the optimal results while ($C_{MM} = 100$) yields poor result because the model fails to converge due to high loss values. The high loss value is because the considerable value of ($C_{MM}$) results in large $l_2$ norm values in the case of MMC. When $C_{MM}$ is set as 100, the resulting accuracy is like picking a class randomly from the ten classes from the dataset.

| $C_{MM}$ values | Resulting accuracy |
|---|---|
| 1 | 92.16% |
| 10 | 92.57% |
| 50 | 68.92% |
| 100 | 10% |

Table 4: Experiment to determine optimal $C_{MM}$ value for use with MMC loss. Models trained on CIFAR10 dataset for 200 epochs using MMC loss.

### 4.7 Feature Representation

Figure 2 represents the feature visualization of MNIST dataset and Figure 3 represents the feature visualization of CIFAR10 dataset. For simplicity, we have used isometric projection on a circle in the case of MMC loss. As evident from the Figure 2 and 3 the sample density is higher in the case of MMC loss.

This fixed untrainable class center in MMC is favored in the MNIST dataset cases where each class is unrelated to the other. While in the case of cifar-10, we want our feature representation to depict both inter-class relation and intra-class relation, which is more favored in SCE loss. In the figure 3(b), the class with orange color represents a cat, and the class with pink color represents a dog. Ideally, we want our feature representation to capture some common relations between dog and cat. They are more similar to each other as compared to other classes like airplane marked in red. In our ideal representation, dog and cat centers should be near to each other than the airplane. This inter-class nature is somewhat captured by SCE loss, not by MMC loss due to fixed untrainable class centers. Figure 3(b) depicts how class centers should be initialized when two class have a common representation. Distance should be minimized between such classes and maximized between other classes.

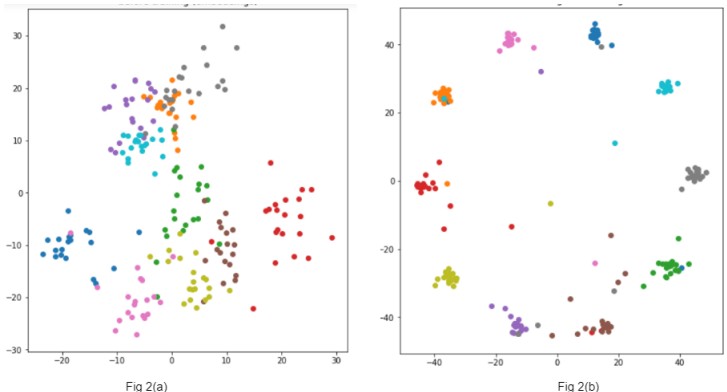

Figure 2: Comparions of Embedding Representation of SCE and MMC for MNIST dataset. Fig 1(a) is for SCE loss and Fig(b) for MMC loss.

Finally, we implement **hierarchical Max-Mahalanobis** (HMMC) loss a variant of MMC loss as mentioned in the supplementary section of the original paper. The authors presented HMMC loss algorithm for datasets like CIFAR100 where each class have multiple subclasses. CIFAR100 has 20 superclass and 5 sub-classes in each superclass. We first generate 20 MMC centers with $C_{MM} = C_1$ and then we generate 5 MMC centers using $C_{MM} = C_2$ where $C_1 >> C_2$. If a label $l$ is the $j^{th}$ class in the $i^{th}$ superclass, then $\mu_l^H = \mu_i + \mu_j$. We experimented with multiple values of $C_1$ and $C_2$. We got poorest results when $C_1$ approach 100. The best range from $C_1$ is between [5, 20] and for $C_2$ is between [0.1, 2]. We were unable to beat the performance of original MMC centers using HMMC centers.

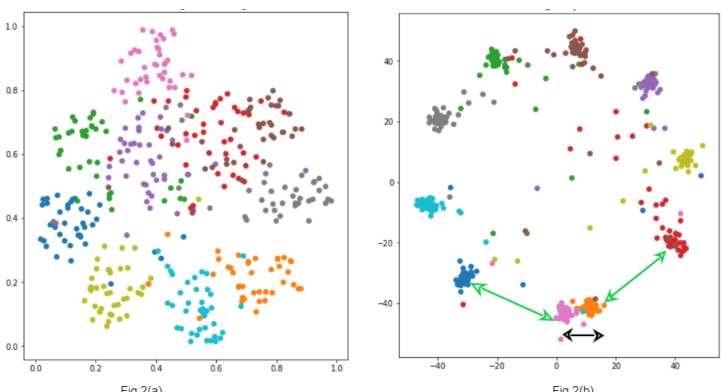

Figure 3: Comparions of Embedding Representation of SCE and MMC for CIFAR dataset. Fig(a) is for SCE loss and Fig(b) for MMC loss.

## 5 Conclusion

Our experiments investigate the validity of the original paper's results, and we find that MMC loss presents as a viable alternative to SCE. Our experiments empirically demonstrate how MMC loss induces high-density regions in the feature space. All our results support the central claims made in the paper. The experiments show that MMC loss leads to reliable robustness under strong adversarial attacks at the cost of little extra computation. The method they have proposed is novel, and their analysis in the paper gives considerable insight into the development of new objective functions.

## 6 Acknowledgements

We want to thank Tianyu Pang, Tsinghua University, for clarifying specific queries throughout this work. We also want to offer our thanks to Prof. Anirban Dasgupta, IIT Gandhinagar, for providing us the computation resources we needed. Moreover, we want to extend our special thanks to Mr. Rachit Chhaya, IIT Gandhinagar, for his valuable feedback on this report.

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
