# OpenReview forum: "Reproducibility Report: Rethinking Softmax Cross-Entropy Loss for Adversarial Robustness"
_ML_Reproducibility_Challenge/2020 — Reject_

### Official Review · AnonReviewer1 · 2021-03-01
**Review summary**

**Rating:** 3
**Confidence:** 5

**Review:**

This paper aims to reproduce the results of (Pang et.al,19) which present Max-Mahalanobis center (MMC) loss to defending the adversarial attack.

I appreciate the empirical efforts of the authors. However, investigating the adversarial robustness does not only require one to reproduce the number of the original paper but also needs more thinking of using a stronger attack to give a 'true evaluation' of the proposed method.

As clearly stated in https://arxiv.org/pdf/2002.08347.pdf,  Pang et.al,19 doesn't succeed in defending stronger adversarial attacks. Then I give a clear reject.

**Familiar With The Original Paper:**

I have read the original paper

**Reproducibility Summary:**

Report has summary

---

### Official Review · AnonReviewer2 · 2021-03-11
**Reasonable reproduction; Could use more depth and attention to the presentation**

**Rating:** 5
**Confidence:** 3

**Review:**

Summary: The summary is clear and highlights the major results of the reproduction.
Scope of Reproducibility: clearly stated and adhered to
Code: re-used author’s code
Communications with Original Authors: Done fairly
Hyper-parameter search: code reuses the author’s hyper-parameters
Ablation study: looks at adversarial trading and optimizer selection
Discussion on results: relatively little discussion about difficulties of reproduction, but does describe difficulties of reproduction
Recommendations: none given
Results beyond paper: additional implementation of methods in Python, some initial results validating MMC’s ability to train models that reject out-of-distribution inputs. However, the experiment is only on a single image, so does not provide much value
Overall organization and clarity: the paper has a lot of clarity and grammatical issues. I’ve documented a few below. I also recommend substantially updating the figure captions so that the figures are relatively self-contained and comprehensive.

2.2 I found this section and the notation presented quite hard to follow. I recommend looking at what math is actually needed to set up remarks 2-4 and then providing more explicit definitions of those terms. For example, l.89 references $N_{k, \bar{k}}$ before the definition. Consider dropping the $D_{k, \bar{k}}$ notation entirely and directly defining the term in l.89 (e.g., “is proportional to the number of points in class k when $\bar{k}$ has the highest prediction amongst other classes. Call this $N_{k, \bar{k}}$”).

2.3 I think this would be clearer if the way to compute the centers (and some of the intuition) were discussed first. For example, before l.105,  consider including some of the content of remark 5.

Minor comments:
L.5 define SCE before using an acronym
L.59 “we then present demerits of MMC loss” — > “we then present the merits of the MMC losses”
L.78 This section has several clarity issues. In particular, consider providing more context on how l.75-77 leads to l.78 as a conclusion.
L.94 “function in during the training procedure” — non-grammatical
L.97 “tend to spread over the space in an sparsely” — non-grammatical
L.127 non-grammatical (NG)
L.131 “also not losing out the high accuracies” — NG
L.133 “supervisor inappropriate supervisory signals” —NG
L.135 “this section roughly” — seems that this comes from the template?
L.201 “we validate the merits of MMC, that is” — run-on sentence


**Familiar With The Original Paper:**

I have not read the original paper

**Reproducibility Summary:**

Report has summary

---

### Decision · Program_Chairs · 2021-03-31

**Decision:**

Reject

**Comment:**

The report does not go above and beyond reproduction of the original paper to reflect upon the results and their larger impacts on the field.